# The Development of an Age-Appropriate Fixed Dose Combination for Tuberculosis Using Physiologically-Based Pharmacokinetic Modeling (PBBM) and Risk Assessment

**DOI:** 10.3390/pharmaceutics16121587

**Published:** 2024-12-12

**Authors:** Xavier J. H. Pepin, Juliana Johansson Soares Medeiros, Livia Deris Prado, Sandra Suarez Sharp

**Affiliations:** 1Simulations Plus, Inc., 42505 10th Street West, Lancaster, CA 93534-7059, USA; sandra.suarez@simulations-plus.com; 2Technological Development Coordination, Instituto de Tecnologia em Fármacos (Farmanguinhos)/Fiocruz, Av. Cmte. Guaranys, 447-Jacarepaguá, Rio de Janeiro 22775-903, Brazil; juliana.soares@fiocruz.br (J.J.S.M.); livia.prado@fiocruz.br (L.D.P.)

**Keywords:** tuberculosis, formulation, pediatrics, PBBM, safety, efficacy, modeling, simulation

## Abstract

**Background/Objectives:** The combination of isoniazid (INH) and rifampicin (RIF) is indicated for the treatment maintenance phase of tuberculosis (TB) in adults and children. In Brazil, there is no current reference listed drug for this indication in children. Farmanguinhos has undertaken the development of an age-appropriate dispersible tablet to be taken with water for all age groups from birth to adolescence. The primary objective of this work was to develop and validate a physiologically-based biopharmaceutics model (PBBM) in GastroPlus^TM^, to link the product’s in vitro performance to the observed pharmacokinetic (PK) data in adults and children. **Methods**: The PBBM was developed based on measured or predicted physico-chemical and biopharmaceutical properties of INH and RIF. The metabolic clearance was specified mechanistically in the gut and liver for both parent drugs and acetyl-isoniazid. The model incorporated formulation related measurements such as dosage form disintegration and dissolution as inputs and was validated using extensive literature as well as in house clinical data. **Results**: The model was used to predict the exposure in children across the targeted dosing regimen for each age group using the new age-appropriate formulation. Probabilistic models of efficacy and safety versus exposure, combined with real world data on children, were utilized to assess drug efficacy and safety in the target populations. **Conclusions**: The model predictions (systemic exposure) along with clinical data from the literature linking systemic exposure to clinical outcomes confirmed that the proposed dispersible pediatric tablet and dosing regimen are anticipated to be as safe and as effective as adult formulations at similar doses.

## 1. Introduction

The development of age-appropriate pediatric formulations is important in order to improve compliance with palatable formulations and provide a safe way to dose patients whose physiology is changing with age. Typical changes from birth to adulthood involve plasma protein concentrations, enzyme or transporter expression, pH and volumes in the GI tract, and organ weights [1,2,3,4,5]. The combination of these maturation processes makes each children’s age group unique in terms of drug absorption, distribution, metabolism, and excretion. In addition, as in adults, the genotypes found in children affect the functionality of some enzymes as is the case for N-acetyl-transferase 2 (NAT2) which is the enzyme responsible for INH metabolism. Mutations on the gene coding for NAT2 can lead to an intermediate acetylator or slow acetylators compared to the wild gene which codes for rapid acetylators. Therefore, the extrapolation of dose requirements for children cannot solely rely on body weight since it could lead to inadequate dosing. The use of model-informed drug development, and in particular physiologically based models, is recognized by scientists and regulatory agencies as a fundamental aspect of designing formulations for children and also waiving clinical testing in certain conditions in this age group for ethical reasons [6,7,8]. These models are also mechanistic enough to comprise aspects such as formulation-specific information, a description of enzyme expression and ontogeny, organ weights, and physiological variations during human growth, making them suitable tools to predict drug exposure in children. PBPK models were previously developed for INH and RIF. Most RIF PBPK models have focused on predicting drug–drug interactions with RIF as the perpetrator. In many of these models, the absorption from the gut lumen of RIF is not mechanistic. In the earlier models developed by Asaumi et al., a first order absorption rate constant was utilized for RIF to predict absorption [9]. In the later models from Asaumi et al., the product of permeability and absorptive surface area from the three first compartments of the gut were proposed to handle RIF absorption, neglecting the impact of drug dissolution [10]. Rasool et al., despite having used a mechanistic platform for RIF absorption, did not explain/consider drug dissolution, and instead optimized the effective RIF permeability in their model to cover drug absorption [11]. Similarly for INH, PBPK models have relied on absorption rate constants to handle absorption from the gut [12]. To the best of the authors’ knowledge, this is the first time that a PBBM is developed for INH and RIF, covering aspects of drug product disintegration, mechanistic dissolution, and pH-dependent drug luminal degradation, together with full PBPK and ontogeny models, i.e., proposing a model able to translate formulation effects from adults to pediatric subjects. In this work, a combination of 50 mg INH and 75 mg RIF in an age-appropriate dispersible tablet, was developed for the maintenance treatment of tuberculosis in children from birth to 10 years of age. Older children and adolescents, when they can swallow a whole tablet, can be dosed using units of the adult fixed dose combination tablets. The purpose of this work was to develop and validate a model to predict the exposure to INH and RIF from the age-appropriate formulation and adult tablet in different age groups from birth to adulthood and verify the safety and efficacy of this treatment in the pediatric population.

## 2. Materials and Methods

### 2.1. Solution Stability and Solubility Tests

The solution stability of RIF and INH was evaluated in acetate buffer, pH 4.5, and phosphate buffer, pH 6.8. Solutions were prepared with combinations of RIF and INH, reflecting the fixed dose combination product’s dose ratio (3:2 for RIF:INH), and kept on an orbital shaker, KS 4000i control (IKA, Staufen, Germany), at 37 °C and 100 rpm. At times 0, 2, 4, 6, 12, and 24 h, aliquots were collected, filtered through a 0.45 µm PTFE membrane, and analyzed using a validated stability-indicating HPLC method. The drug degradation over time was determined by measuring the reduction in each drug peak area. The data were expressed as percent intact drug over time to calculate the first order degradation rate constant. For solubility determination, suspensions of the drugs in combination, prepared with concentrations corresponding to the drug product ratio (3:2 RIF:INH), were prepared in in the same media as previously described for degradation. The suspensions were agitated on an orbital shaker at 100 rpm. Aliquots were collected at 2, 4, 6, 12, and 24 h, filtered through a 0.45 µm PTFE membrane, and quantified using the HPLC methods.

HPLC analyses were conducted to quantify INH and RIF using the same stability-indicating method, with selectivity for both drugs and their degradation products. The analyses were carried out on a LC 20AT T system (Shimadzu, Kyoto, Japan) coupled with a C18 column (5 µm, 150 × 4.6 mm) and a diode array detector set at 260 nm. The mobile phase A consisted of potassium phosphate pH 6.6/ethanol in a 94:6 ratio, and mobile phase B was a mixture of methanol/acetonitrile in a 70:30 ratio. The flow rate was maintained at 1.0 mL/min, under the gradient elution procedure, and the column temperature was set to 30 °C. The solvent gradient programming was from 100% to 74% A over 0.5 to 0.6 min, held at 74% A until 0.9 min, from 74% to 70% A over 1.2 min, held at 70% A until 1.5 min, from 70% to 46% A over 2.0 min, held at 46% A until 2.4 min, from 46% to 4% A over 7.2 min, and then returned to 100% A at 9 min, and remaining at 100% A until 18 min.

### 2.2. Dissolution Profiles

The dissolution test was performed in six vessels using a VK7010 dissolution tester (Varian, Palo Alto, CA, USA). Dissolution profiles were obtained using USP Apparatus II at 50 rpm, and 900 mL of HCl 0.1 N at 37 °C. Aliquots of 10.0 mL were withdrawn at 5, 10, 15, 30, 45, and 90 min, without medium replacement. All samples were passed through a 35 µm ultra-high molecular weight polyethylene filter and after appropriate dilution, the samples were quantified. Isoniazid was determined using a LC 20AT chromatographic system (Shimadzu, Kyoto, Japan) with an ACE C18 (5 µm, 150 × 4.6 mm) column and a µBondapack C18 (10 µm, 125 Å) guard column and detection at 254 nm. The mobile phase consisted of 0.08 M dibasic potassium phosphate and 0.59 M monobasic potassium phosphate in isocratic mode, with a flow rate of 1.5 mL/min. Rifampicin was quantified using a UV/Vis spectrophotometer 1900i (Shimadzu, Kyoto, Japan) with a 0.2 cm optical path length cell at a wavelength of 475 nm. Concentration was determined from the calibration curve of the standard solution.

### 2.3. Modeling Strategy

The modeling strategy adopted for this work is presented in Figure 1.

The PBPK model was built in GastroPlus^®^ v9.8.3 and the ADMET Predictor ^TM^ v10.3 was used in addition to predict some missing physico-chemical and biopharmaceutical properties of INH and RIF. The metabolic clearance was specified mechanistically based on in vitro data for intrinsic clearance and in silico predictions of the Michaelis–Menten constant (K_m_) values for the known enzyme isoforms involved in the drug and downstream metabolite metabolism. All the pediatric simulations were made using built-in ontogeny models for the relevant enzymes. These models adapt the expression level of the enzymes in the PBPK model depending on the age and maturation of the enzyme. For NAT2, the ontogeny model was absent from the software and was integrated. In addition, the genetic polymorphism which largely impacted the drug clearance for INH was integrated into the model. This ontogeny model is described in the literature, and the main points are shown in the Appendix A [13]. The mechanistic absorption model developed for adult subjects was extended to simulate drug absorption and pharmacokinetics in pediatric populations. The ACAT model in GastroPlus automatically adjusted the size (length and diameter) of each gut compartment based on body weight for different age groups. Changes in stomach volume and pH with age were also incorporated into the model using data from de Zwart et al. [14]. However, due to limited information, the gastric emptying time in pediatric populations was assumed to be the same as in adults. Similarly, data on small intestine transit time in pediatric populations were not available, so the program calculated total small intestinal transit times in pediatric populations based on reported oro-fecal transit times and gastric emptying values, using the typical body weights for the different age groups. Transit times for individual small intestine compartments were estimated using constant fluid flow proportional to relative transit times in adults. Colonic transit times were calculated based on polynomial equations fitted to the literature data [15]. Since the in vitro dissolution for clinical drug products is mechanistically integrated for INH and RIF (Section 2.7 and Appendix A), the in vivo dissolution will be different for all the simulated subjects based on the impact of volume, transit time, pH, dose, and bile salt concentration, according to the physiological changes related to age in each population. PBBM is a fit-for-purpose modeling approach that, from a regulatory submission perspective, involves constructing a dissolution safe space via an IVIVR or IVIVC to support drug product quality. A key step in PBBM development is incorporating in vitro dissolution as an input followed by modeling in vivo dissolution with a mechanistic understanding. In the present work, since the in vivo dissolution is mechanistic, the model is being referenced to as a PBBM, even if its purpose is not strictly to define quality specifications for the drug product, but to predict the impact of formulation differences in adults and children. A dissolution safe space was developed for INH using this PBBM as a separate exercise as shown by Pepin et al. [13].

For pediatric simulations, the PBPK models were adjusted using age-dependent physiological parameters integrated into the population estimates for the age-related physiology (PEAR™) module in GastroPlus 9.8.2. These models included age-specific values for tissue compositions, blood flows, tissue sizes, hematocrit, extracellular tissue water, total body water, and plasma protein concentration reported in the literature. Total body height, weight, and most tissue sizes were obtained from ICRP publication 23 [16] and other relevant literature sources. Tissue-specific blood flows in mL/s/mL-tissue were extracted from publications, and age-dependent tissue composition data were collected from the available literature [17,18]. Further details can be found in the program’s manual.

The PBPK model also accounted for changes in renal function with age. Equations for calculating age-dependent glomerular filtration rate (GFR) were employed for children aged 0 to 6 years and from 6 years to adults, based on Kearns et al. [19] and Stevens et al. [20], respectively.

GastroPlus also accounts for differences in the fraction unbound in plasma (Fup) and the blood/plasma concentration ratio (Rbp) between children and adults due to different levels of protein and hematocrit, respectively. The Fup scaling is based on the previously published [21] Equation (1), and assumes that the input experimental percent unbound in plasma is representative of nonspecific drug–protein binding in adult plasma:(1)$${\mathrm{Fup}}_{\mathrm{ped}}=\frac{1}{1+\frac{P_{\mathrm{ped}}}{P_{\mathrm{adult}}}\times \frac{1-{\mathrm{Fup}}_{\mathrm{adult}}}{{\mathrm{Fup}}_{\mathrm{adult}}}}$$
where P_ped_ and P_adult_ represent protein concentrations in pediatric and adult plasma, respectively, and Fup_ped_ and Fup_adult_ represent fraction unbound in pediatric and adult plasma, respectively. The ratio of pediatric to adult plasma protein (P_ped_/P_adult_) is based on the ontogeny of the two major drug-binding proteins in plasma: α1-acid glycoprotein (AAG) and albumin. Infant Rbp scaling utilizes Equation (2) assuming that the input experimental Rbp value represents binding to red blood cells in adult blood (hematocrit = 0.45).
(2)$${\mathrm{Rbp}}_{\mathrm{ped}}=\frac{{\mathrm{Hct}}_{\mathrm{ped}}}{0.45}\times \left({\mathrm{Rbp}}_{\mathrm{adult}\;}-\left(1-0.45\right)+\left(1-{\mathrm{Hct}}_{\mathrm{ped}}\right)\right)$$
where Hct_ped_ represents hematocrit (expressed as a fraction) in pediatric blood; Rbp_ped_ and Rbp_adult_ represent blood/plasma concentration ratios in pediatric and adult blood, respectively. Whole body physiology was created to match closely the mean population parameters (age, body weight, and body mass index) of the population used in the clinical trials reported.

### 2.4. Modeling Assumptions

The default ACAT models for human fasted or fed physiology, depending on the prandial state, and including NAT2 expression for INH are described in the Appendix A. For Ac-INH or RIF, default CYP1A2, CES2, and CYP3A4 expression levels were utilized. The default Optimized log D Model SA/V 6.1 was applied to calculate absorption scale factors. This model adjusts the absorption scale factors for changes in permeability due to ionization, villi density, and possibly tight junction gap (if paracellular absorption is not treated as a separate process) among the compartments. The coefficients C1-C4 are set to default values, which have been optimized to provide the best fit to the observed data for a series of drugs with known human effective permeability (P_eff_).

For coated FDC tablets, the stomach transit time was assumed to be that of the predicted in vivo disintegration time, i.e., on average 0.5 h. In study STPh71/10, the in vitro pharmacopeia disintegration of the test was 20% longer than that of the reference (8.6 min for the test compared to 7.1 min for the ref using pharmacopeial disintegration testing). The literature reports that the in vitro pharmacopeia disintegration test is not representative of the in vivo disintegration and that lower agitation or fluid velocity exists in vivo [22,23,24]. On average, in vivo disintegration takes 4 times longer than in vitro disintegration and the RSD of in vivo disintegration is approximately 60% [22]. This means that both combination tablets in study STPh71/10 could take 34 min for the test on average (0.57 h) and 29 min for the reference (0.48 h) to disintegrate. These times are used for stomach emptying on top of drug product dissolution differences since it was reported in the literature that tablets do not leave the stomach until full disintegration [25].

For RIF, since this is a BCS class 2 product which is solubility limited, the volume in the small intestine was reduced to 20% and 2% in the colon. These adjustments reflect the volume of water available in the lumen and in the mucus for the upper part of the GI tract and available in the lumen for the lower part of the GI tract, reflecting the measurements obtained by MRI in humans [26,27]. In addition, the option of restricting further the water in the small intestine to 7.5% for children is tested on measured oral PK profiles during model validation.

### 2.5. Criteria for Model Validation

The following two model prediction performance indicators are calculated for the PK parameters or concentration–time profiles where relevant for model validation.

The average fold error (AFE) is defined by Equation (3).
(3)$$AFE=10^{\frac{1}{n}\sum \log\frac{Pred_{i}}{Obs_{i}}}$$

The AFE is an indicator of the prediction bias. A method that predicted all observed values with no bias would have a value of 1; under-predictions are shown by an AFE below 1 and over-predictions by AFE values above 1. AFE values vary between 0 and infinity; in general, a prediction may be considered satisfactory if the AFE is between 0.8 and 1.25, passable if the AFE is within [0.5–0.8] or [1.25–2], and poor if the AFE is within [0–0.5] or above 2. For this work, a satisfactory AFE is needed.

The average absolute prediction error (PE%) is defined by Equation (4).
(4)$$PE\left(\%\right)=Geomean\left(\left|\frac{Pred_{i}-Obs_{i}}{Obs_{i}}\right|\right)\times 100$$

PE is a measurement of prediction scaled to percentage units, which makes it easier to understand. It is very close quantitatively to (AAFE − 1) × 100. A model is considered satisfactory when the PE is less than 25%, passable if the PE is between 25 and 50%, and poor if the PE is ≥50%.

For INH, C_max_ and AUC_0-t_ in adults and children are used to predict model performance using the above criteria. For RIF, C_max_ and AUC_0-t_ in adults are used to predict model performance using the above criteria. For children and RIF, due to sparse sampling, the actual concentrations for each time point are used for each study and age group to calculate the model prediction performance using the above criteria.

### 2.6. Integration of Disintegration in the PBBM

The literature reports that the in vitro pharmacopeia disintegration test is not representative of the in vivo disintegration and that lower agitation or fluid velocity exists in vivo [22,23,24]. On average, in vivo disintegration takes 4 times longer than in vitro disintegration and the RSD of in vivo disintegration is approximately 60% [22]. The in vitro disintegration observed for the FDC tablets was therefore multiplied by 4 to estimate the gastric emptying time of these tablets in the fasted state, since because of their size, they would not empty from the stomach prior to full disintegration. For the age appropriate formulation, the gastric emptying was either left at the default value for tablets in GastroPlus (0.25 h) or set to the default value of suspensions (0.1 h), which typically empty in the fasted state with the administration fluid [27].

### 2.7. Integration of Dissolution in the PBBM

For model validation and use, a mechanistic dissolution model was used to integrate the measured in vitro dissolution of tablets in the model. For INH, the Z-factor introduced by Takano [28] was utilized to fit the dissolution data obtained for the literature and clinical batches. This approach was validated using the INH dissolution data reported by Gelber et al. [29] on different INH tablet formulations [13], and the Z-factor was applied to INH dissolution from the age-appropriate fixed dose combinations tested in this work (Appendix A). The Z-factor was also utilized to fit the dissolution data obtained for the literature and clinical batches, except for the study reported by Agrawal et al. [30,31], where an obvious effect of agitation was seen in the dissolution data attesting to the presence of coning. Instead, for the drug products reported by these authors, a P-PSD HD model was utilized [32] to fit the in vitro dissolution, since this model integrated the effect of excipients on the formulation coning during dissolution. The fitting of dissolution data from Agrawal et al. is shown in the Appendix A.

### 2.8. Dosage Forms

The age-appropriate fixed dose combination tablet comprises 50 mg INH and 75 mg RIF. This tablet is dispersible in water and therefore is adapted to dosing pediatric patients from birth by using dispersion in water. The formulation comprises typical immediate release excipients, all generally recognized as safe, and sucralose for taste masking.

The Farmanguinhos INH tablet described in study STPH08/19 is an immediate release tablet containing mannitol, pregelatinized starch, corn starch, microcrystalline cellulose, magnesium stearate. The product is obtained by aqueous granulation.

The Farmanguinhos fixed dose adult tablets described in study STPH71/10 are coated tablets comprising 150 mg INH and 300 mg RIF. The tablet cores are manufactured by wet granulation and contain microcrystalline cellulose, partially pregelatinized corn starch, croscarmellose sodium, povidone, sodium lauryl sulfate, colloidal silicon dioxide, and magnesium stearate. The coating is aqueous, containing polyvinyl alcohol, titanium dioxide, talc, soy lecithin, xanthan gum and ponceau red 4R aluminum lake dye.

The type and amount of excipients in the test and reference formulations analyzed in this work, together with the BCS classes of INH and RIF (BCS class 1 and 2, respectively), do not lead to anticipated differences in their in vitro and in vivo performance that would not be captured by disintegration or dissolution tests.

### 2.9. Clinical Studies for Model Validation

The clinical trials for model validation were primarily taken from the literature and are described in the Appendix A. Two internal studies, performed by Farmanguinhos to support the development of a 100 mg INH tablet (Study STPh08/19) and a coated fixed dose combination tablet for adult maintenance treatment of TB, comprising 150 mg INH and 300 mg RIF (Study STPh71/10), are described hereafter.

Study STPh08/19 was conducted in Brazil on 42 healthy volunteers. The 100 mg INH tablets were administered after a 10 h fast. The test tablet batch was FIOCRUZ/RJ batch 18060194; the reference tablet batch was Isozid^®^ Riemser Pharma GmbH batch 002017. Drugs were administered with 200 mL water and no food was allowed until 4 h after dosing. No water was allowed until 2 h after dosing. Individual PK profiles for INH were measured for the reference and test formulations.

Study STPh71/10 was conducted in Brazil on 28 healthy volunteers. The study was a randomized 2 period crossover bioequivalence study with a 1 week washout. The drug products were fixed dose combination tablets comprising 150 mg INH and 300 mg RIF. The drug products were administered with 200 mL water. No water and food were allowed until 2 h and 4 h after drug administration, respectively. The test drug product was FIOCRUZ/RJ batch 09060664. The reference batch was Rifinah^®^ Sanofi Batch A9362. Oral individual PK profiles for INH and RIF were measured.

Overall, for the PBBM validation, there were for INH, 10 adult clinical trials, where INH was administered fasted or fed with different types of meals with doses ranging from 150 to 1000 mg. In addition, for INH, there were 6 pediatric studies with doses ranging from 5 to 25 mg/kg. The mean pediatric patient age in these studies ranged from 6.6 months to 9 years. Some of these clinical studies reported INH PK in pure acetylator types. When the acetylator type was not mentioned, genotype distribution was estimated from the terminal INH half-life in individual subjects when available, or the genotype distribution was based on population studies. Some of these clinical studies reported Ac-INH PK as well, which was used to assess the model performance in the prediction of INH metabolism in the different acetylator types.

In addition, for RIF, the model validation relied on 7 adult clinical trials (26 different scenarios) in the fasted or fed prandial state, with or without acid reducing agents (ARAs). The adult doses for RIF ranged from 2 to 35 mg/kg. There were also 9 pediatric clinical trials (18 clinical scenarios) for the PBBM validation. The mean age for pediatric patients where RIF PK was measured was 1 day to 18 years. The details of these clinical trials and their reference are shown in the Appendix A.

### 2.10. Model Application for Adult and Pediatric Simulations

#### 2.10.1. Adult Simulations

The impact of altered gastric emptying due to the type of formulation (dispersible or coated tablet) was assayed through the conduct of population cross-over simulations. The average gastric emptying for the dispersible tablet was set to 0.1 h, which is typical of that for a suspension in GastroPlus (default), whereas the gastric emptying for the whole tablet was set to 0.5 h on average, which would be representative of delayed emptying, due to more prolonged in vivo disintegration of the tablet. The dissolution was governed by the extreme Z-factors calculated on the dispersible or whole tablets. Indeed, formulations displaying the biggest difference in terms of Z-factor, namely, dispersible tablet batch 2111EX054 (INH Z-factor = 4.5 × 10^−5^ mL/mg/s) and tablet batch NRT9104 RIF Z-factor = 2.3 × 10^−3^ mL/mg/s), were used as the fasted batches, whereas the coated batch 09060664 (INH Z-factor = 1.13 × 10^−5^ mL/mg/s) and coated batch A9362 (RIF Z-factor = 6.63 × 10^−4^ mL/mg/s) were used as the slowest dissolving batches. Cross-over simulations were conducted in adults in the fasted state at a 300 mg dose for INH and 600 mg dose for RIF.

#### 2.10.2. Pediatric Simulations

The model was used to predict the exposure from the dispersible tablet comprising 50 mg INH and 75 mg RIF described in Section 2.8, the dissolution of which is fitted with the Z-factors shown in the Appendix A. The target administration schedule for children aged from birth to 11 years was based on the body weight recommendations for dosing of INH and RIF. The body weight vs. age relationships were calculated from the WHO standards (https://www.who.int/tools/child-growth-standards/standards/weight-for-age, accessed on 19 November 2024) and data published by Valentin et al. [33], which translate into the proposed dosing regimen in Table 1.

Using Table 1, 5 age groups can be defined in children for dosing of the dispersible tablets comprising 75 mg RIF and 50 mg INH. These groups are defined below:Group 1: 0–1 year. This group should receive 1 dispersible tablet.Group 2: 1–3 years. This group should receive 2 dispersible tablets.Group 3: 3–5 years. This group should receive 3 dispersible tablets.Group 4: 5–8 years. This group should receive 4 dispersible tablets.Group 5: 8–11 years. This group should receive 6 dispersible tablets.

In addition, as a comparison with the coated adult tablets, 3 groups are defined in adolescents and adults: group 6: 11–13 years, group 7: 13–18 years, and group 8: 18–25 years. These groups served as comparisons and received a dose of 2 whole adult tablets (300 mg INH + 600 mg RIF). The simulations were conducted with a variable gastric emptying between and within subjects in the fasted state. The baseline gastric emptying for the dispersed tablet was set to 0.25 h, and 0.5 h for the whole tablet. For INH simulations, 75 children were simulated in each age group, 25 RA, 25 IA, and 25 SA for INH. For RIF simulations, 25 children were simulated in each age group.

## 3. Results

### 3.1. Physicochemical and Biopharmaceutical Properties

The physicochemical and biopharmaceutical properties for isoniazid (INH), acetyl-isoniazid (Ac-INH), and rifampicin (RIF) were defined using a combination of in silico estimates from the ADMET Predictor module that were based on the chemical structure, along with in vitro and in vivo data obtained from the literature. Figure 2 shows the structure of INH, Ac-INH, and RIF. Table 2 summarizes the parameter values used for the INH PBPK model, Table 3 summarizes the parameter values for the Ac-INH PBPK model, and Table 4 summarizes the parameter values used for the RIF PBPK model.

The detailed explanations of the solubility vs. pH, Log D vs. pH, drug degradation rate in the lumen, precipitation time, distribution and clearance models, and drug permeability are shown in the Appendix A. The dissolution data of batches that were used for the prediction of human and pediatric PK profiles for reference and test batches are shown in the Appendix A together with the resulting fit with the mechanistic dissolution model.

### 3.2. Model Validation

The PBBM validation for INH in adults was previously reported [13]. The profiles are shown in the Appendix A for adults and pediatric patients. The PK profile predictions for adults and pediatric patients with the PBBM for RIF are shown in the Appendix A. Due to the sparse sampling for pediatric profiles, the predicted concentrations at measured time points were used instead of the PK parameters to calculate prediction performance indicators for the pediatric population. For adults, AUC and C_max_ are utilized. The overall PBBM performance to predict the exposure to INH and RIF in adult and pediatric subjects and patients is shown in Figure 3. The detailed calculation of model prediction performance indicators is shown in the Appendix A and a summary is provided in Table 5.

Overall, the model prediction performance for INH passes the predefined criteria for model validation and is considered acceptable. Nevertheless, the predictions are within the measured PK variability, especially in pediatric subjects (Appendix A). Overall, the model prediction performance for RIF passes the predefined criteria for adults and is considered acceptable. For pediatric subjects, the AFE is satisfactory whilst the PE is passable. The PE was calculated on each time point of the PK profile, hence the higher error prediction compared to the PE calculation on C_max_. Indeed, variation in gastric emptying, and in vivo tablet disintegration can influence the concentration–time profiles. The slight over-estimation of plasma concentrations for the pediatric patients could be related to impaired gut membrane permeability in patients with TB [46] and also to the reported PK data which are rarely on naïve subjects. In addition, since RIF induces its own metabolism, the clearance is increasing over time which would reduce the observed exposure. Nevertheless, the predictions made in naïve subjects are within the measured PK variability for pediatric subjects and the population simulations at steady-state for RIF comply with the reported exposure values in the literature for adults and children (Appendix A). The model is therefore considered validated for adults and children.

### 3.3. Examination of Individual Profiles

In study STPh71/10, adult fixed dose combination tablets comprising 150 mg INH and 300 RIF were administered in the fasted state in a cross-over bioequivalence study. Both test and reference formulations had similar in vitro characteristics in terms of disintegration (8.6 min for the test compared to 7.1 min for the ref using pharmacopeial disintegration testing) or dissolution. The examination of individual profiles reveals multiple absorption peaks for INH and RIF which are synchronized. These multiple absorption phases attest to the presence of multi-phasic gastric emptying in 50% of the subjects in the study using the solid oral fixed dose combination. Multiple peaking has been previously reported for a variety of drugs in the fasted state and appears to be related to the fasted state motility and random location of the dosage forms in the stomach [47,48,49,50,51,52,53]. Multi-phasic gastric emptying has a pronounced effect on the observed C_max_ on the AUC for this combination product, since a longer stomach residence of INH and RIF will increase the degradation of these drugs in solution. Since the pediatric formulation is a dispersible tablet to be administered with water, the difference in gastric emptying time compared to a solid oral combination product is anticipated to be large. The detailed analysis of these multiple peaks and conclusions regarding risks related to future human studies are shown in the Appendix A.

### 3.4. Model Application

#### 3.4.1. Model Application to Adult Simulations

Results for model application to virtual cross over studies in adults comparing the pediatric dispersible formulation (test) to the adult fixed dose combination tablets for 300 mg INH and 600 mg RIF are shown in Table 6.

Overall, the INH C_max_ GMR for the dispersible tablet versus the whole tablet ranges from 108% to 115% with a 90% CI, which is close to the upper acceptance limit for BE. Although for these three INH trials there was no failure of virtual BE, there is a chance more VBE trials may fail due to INH C_max_. As mentioned previously on observed PK profiles observed in study STPh71/10, RIF is less sensitive to variations in gastric emptying phases. Table 6 shows that the test dispersible tablet would lead to a 2% higher RIF C_max_ compared to the whole tablet. As seen in previous BE studies, the C_max_ for INH is more sensitive to gastric emptying than the C_max_ of RIF.

#### 3.4.2. Model Application to Pediatric Simulations

The predicted AUC and C_max_ for all these population simulations are shown in the Appendix A and are in good agreement with the measured values from the literature. Namely, the INH-predicted Cmax and AUC values are in accordance with the reported literature values at equivalent doses [54,55,56,57,58,59,60,61]. In addition, the predicted RIF C_max_ and AUC values comply with the values reported in the literature at equivalent doses [55,59,62,63,64,65,66,67]. The INH and RIF C_max_ and AUC predicted as a function of age when children are dosed according to the schedule shown in Table 1 are shown in Figure 4 and Figure 5, respectively.

It is interesting to note that the faster gastric emptying hypothesized for the dispersible tablets compared to the coated adult FDC tablets translates, for INH, to higher C_max_ values for the age groups receiving the dispersible tablets compared to the older children dosed with the adult tablet. The lower exposure predicted for whole tablets compared to dispersible tablets in this analysis for RIF is related to prolonged gastric residence, which induces higher degradation. There is a marked effect of genetic polymorphism on NAT2 on the exposure to INH across the two formulations.

## 4. Discussion

### 4.1. Isoniazid

The pharmacokinetic profiles of INH show multiple peaks [29,70], which lead to variability within subject, especially regarding C_max_. This variability can be up to 40% as shown by Peloquin et al. [71], and confirmed by internal studies from Farmanguinhos. The PK of INH does not seem to depend on stomach pH [71], but the food type and co-administration with other anti-TB drugs will affect the PK of INH. Food is shown to reduce the C_max_, as expected from the effect on gastric emptying. However, the type of food will impact the C_max_ and AUC of INH. Carbohydrate-rich diets tend to lead to a reduction in exposure compared to high-fat meals [70,71,72,73]. Devani et al. have shown that reducing sugars react with INH and lead to the formation of hydrazones [74]. Using the degradation rates measured for various reducing sugars and INH at 37 °C, it is possible to predict the loss of AUC corresponding to the administration of carbohydrate-rich diets as reported by Pepin et al. [13]. It is recommended not to use reducing sugars in the formulation or vehicles used to administer INH-containing formulations to avoid degradation of the drug. Based on the sensitivity analyses, the most influential parameter on the C_max_ (and also the AUC due to luminal degradation and first-pass gut extraction) is the gastric transit time. The INH formulation type is expected to impact the frequency of multiple phases in the gastric emptying: A dispersed formulation is anticipated to empty with the fluid in the fasted state with a gastric emptying half-time of about 6–7 min [27], whilst solid formulations will stay for prolonged periods and variable times in the stomach while they disintegrate [75]. For dispersible tablets and adult FDC tablets, the dissolution was mechanistically integrated in the PBBM, and the in vitro observed disintegration time was scaled for in vivo disintegration to inform the gastric emptying time. The difference of 0.48 h for the reference tablet and 0.57 h for the test tablet stomach transit time in clinical study STPH71/10 was enough to explain the observed C_max_ and AUC difference for INH, which is related to the small difference in first-pass gut extraction between the two tablets. It was hypothesized that a dispersible tablet would empty with the administered water, i.e., with the default gastric emptying rate of suspensions (0.1–0.25 h). Conversely, INH in vitro and in vivo dissolution is shown not to be rate limiting for drug absorption [13]. In addition, the time spent in the stomach for a combination product also leads to further product elimination by the degradation of INH and RIF and/or by a dilution of the drug in front of the absorptive surface and higher first-pass gut extraction.

This combination of factors may lead to failing the BE study in adults when comparing the dispersible pediatric tablet with the adult FDC tablet, especially on INH C_max_. The comparison of RIF and INH PK could help isolate the subjects for which the gastric emptying is limiting drug absorption. Based on the estimate of RA, IA, and SA prevalence in the Brazilian population of 0.11, 0.55, and 0.34, respectively, [76], and based on the simulations for C_max_ GMR in RA, IA, and SA found in Table 6, the estimated C_max_ GMR for a dispersible tablet versus a whole tablet reference is estimated at 113.4%. Assuming a 40% within-subject variability for INH C_max_ in FDC coated tablets, the estimated size of a two-way cross-over BE study to pass BE on C_max_ is 91 subjects, whilst it is 44 subjects for a four-way fully replicated crossover BE study, using the reference scaled average BE approach [77].

In children, this mechanism of variable and different gastric emptying was also considered in the simulation and shown not to impact the exposure outside of what was observed in the clinic for pediatric subjects. In many pediatric PK studies reported in the literature, the tablets are crushed, or the dose of INH is solubilized from drug powder in water, and therefore the stomach emptying is fast.

Simulations conducted for each age group according to the dosing schedule of Table 1 for INH have shown that the AUC is well-predicted compared to the literature values and that C_max_ may be higher than reported. However, due to the lack of rich PK data in pediatric subjects, this higher C_max_ may be due to the lack of early sampling since C_max_ is frequently missed even with frequent sampling. Overall, there is not a significant difference between the measured and predicted C_max_ when one considers the standard deviation of the measurements (Appendix A). There is a large variability in INH C_max_ and AUC, especially for group 1, which groups children from birth to 1 year old. The main drivers of these differences are the dose expressed in mg/kg and the acetylator status. In addition, in this age group, the ontogeny of the NAT2 and liver growth also play an important role.

Overall, there are no differences in AUC vs. age using the dosing schedule proposed in Table 1. The risk of developing a DILI (elevation of liver function markers) based on INH exposure alone and depending on the prevalence of acetylator types in the Brazilian population would range from 13.5% to 18.7%, which is equivalent to what is observed in adults [69] and equivalent to that reported in children [78]. It is recommended that the pediatric population be monitored for liver function. The risk of developing jaundice or actual liver failure is relatively low based on equivalent or higher dose reported cases in pediatric patients and will depend on the co-morbidities. For pulmonary TB, jaundice incidence due to INH treatment is of 0.8% and rises to 10% in meningitis TB, although co-administered drugs and co-morbidities in the case of meningitis TB could play important aggravating factors. The risk of liver failure (signs of drug-induced liver injury) is low, with an incidence of 0.4–0.6% at doses equivalent to or higher than 10 mg/kg.

### 4.2. Rifampicin

Rifampicin also displayed multiple peaks in the pharmacokinetic profiles when present in an FDC; these peaks are aligned with the INH peaks and this synchronicity attests to the impact of stomach emptying. These peaks were observed, for example, during study STPH71/10 (Appendix A) and are also reported in the literature [79].

Rifampicin induces its metabolism during chronic administration. The effect of repeated administration of RIF was predicted at steady state exposure, with an increase of 72% V_max_ for all enzymes present in the gut or liver. In addition, RIF is sensitive to pH degradation and its association with INH also leads to faster degradation in the stomach. Gastric retention in combination with fast drug product dissolution can lead to reduced drug exposure. The exposure to RIF was also found to be solubility and dissolution rate limited in the human intestine. The predictions of the effect of food or PPI on the exposure to RIF is well captured, which demonstrates that the hypothesis on the volume available for dissolution are verified and that the effect of pH on solubility and the combined effect of pH and transit time on degradation are well captured by the model (Figure 6). As a consequence, it is recommended not to use vehicles of pH 2–5 to administer RIF-containing products, since this may increase the drug degradation in the stomach.

In terms of efficacy prediction based on RIF exposure, the risk of a 2-month positive sputum is low at 4.6% and the risk of therapy failure after 2 years is of 13.1%. Both these values are twice lower than observed by Pasipanodya et al. in adults treated with rifampin 10.90 mg/kg, isoniazid 6.52 mg/kg, pyrazinamide 35.71 mg/kg, and ethambutol 24.62 mg/kg [68]. It can be concluded that the RIF exposure resulting from dosing the dispersible tablet to children according to the schedule proposed in Table 1 is anticipated to be more effective than in adults. In terms of DILI and AKI, exposure to RIF is safe with lower risk of DILI compared to INH and lower risk of AKI compared to adults as corroborated by the literature values in children [78,81].

### 4.3. Combined View on Safety and Efficacy

#### 4.3.1. Efficacy of INH and RIF

INH inhibits mycobacterium tuberculosis membrane formation whereas RIF inhibits bacterial RNA polymerase. Roy et al. report that the MIC for INH is 0.025–0.05 μg/mL [58]. Donald et al. showed that the early bactericidal activity EBA_90_ is achieved for an AUC_inf_ higher or equal to 10.52 ± 3.69 μg.h/mL or an INH C_2H_ of 2.19 ± 0.68 μg/mL [82,83]. Similarly, an AUC0-24 threshold of 10.52 is reported by Kiser et al. [57], who also add a minimum C_max_ requirement of 3 μg/mL for efficacy for INH. Regarding RIF, Pasipanodya et al. identified a peak concentration threshold of 6.6 μg/mL to be predictive of 2-month sputum conversion in a study of 142 TB patients [68]. Zheng et al. evaluated the distribution of MIC for INH and RIF in *M. tuberculosis* isolates from 168 and 52 patients with drug-susceptible tuberculosis in development cohort (A) and validation cohort (B), respectively [69]. The most sensitive *M. tuberculosis* strains show MIC for RIF at 0.12 μg/mL. At this concentration, 85% of the strains are inhibited on average. At RIF concentrations of 0.25 μg/mL, 93% of the strains are inhibited. For INH, a concentration of 0.025 μg/mL only inhibits 18–25% of the most sensitive strains, whilst a concentration of 0.05 μg/mL inhibits 93% of the strains. Zheng et al. concluded that the anti-TB treatment success after a 2-month treatment in adults with average dose combinations of 10 mg/kg RIF, 6 mg/kg INH, 14 mg/kg ETH, and 27 mg/kg PYR was conditioned to a minimum RIF AUC/MIC ratio equal to 435 [69]. The safety and efficacy of RIF vs. INH was evaluated by Diallo et al. in 6–14-year-old children using average doses of 17 mg/kg and 10 mg/kg, respectively [81].

In conclusion, in children with latent tuberculosis, a regimen of 4 months of Rifampin had better rates of completion than 9 months of isoniazid, with similar safety profiles in the two trial groups. Rifampin has the advantage of being a single-drug regimen with existing palatable formulations for children. Both treatments had similar efficacies. In terms of efficacy, and following dosing of the dispersible FDC tablet to children using the regimen of Table 1, the exposure is anticipated to be effective for INH since all subjects treated with dispersible tablets are anticipated to show effective concentrations (Figure 4). For RIF, the anticipated C_max_ would be above the 6.6 μg/mL threshold in 80% of the population, this would translate to a risk of positive 2-month sputum culture in 4.6% of the population (Table 7). This risk is less than the 11% of positive 2-month sputum culture observed by Pasipanodya et al. in adults treated with rifampin 10.90 mg/kg, isoniazid 6.52 mg/kg, pyrazinamide 35.71 mg/kg, and ethambutol 24.62 mg/kg [68]. When the RIF AUC is considered, the probability to have a poor 2-year therapy outcome is estimated at 13.1% (Table 8).

This risk is less than the 25% of poor 2-year therapy outcome observed by Pasipanodya et al. in adults treated with rifampin 10.90 mg/kg, isoniazid 6.52 mg/kg, pyrazinamide 35.71 mg/kg, and ethambutol 24.62 mg/kg [68]. It is important to note that the exposure to the tablet was certainly underestimated due to the options of prolonged gastric residence and drug degradation taken as worst-case scenarios in these simulations. Note that since the coated tablet is already available on the market and has demonstrated efficacy in patients, this risk is minimized.

#### 4.3.2. Safety of INH and RIF

Animal studies have demonstrated that the hepatotoxicity of isoniazid is caused by acetyl-hydrazine which is formed by hydrolysis of the major INH metabolite acetyl-isoniazid. The higher incidence of INH-related hepatotoxicity in rapid acetylators has been attributed to their higher rate of formation of acetyl-hydrazine from acetyl-isoniazid [84]. Zheng et al. reported that DILI (defined as alanine transaminase elevations above five times the upper limit of normal (ULN) or above three times the ULN with total bilirubin above two times the ULN after 2 months of treatment) is observed in 32% of patients for an INH AUC > 21.78 μg.h/mL (no DILI below this threshold for 91% of the patients), whilst an AKI (defined as an increase in serum creatinine of more than 26.52 μmol/L or a 1.5-fold increase from baseline level within 2 weeks of TB treatment) is observed in 58% of patients for a RIF AUC > 82.01 μg.h/mL (no AKI in 92% of the patients below this threshold) [69]. Satyaraddi et al. evaluated the relationship between the incidence of DILI and RIF concentration, in combination with INH and PYR treatment [85]. They found that a C_max_ of 12.2 ± 2.3 μg/mL or an AUC_0–4h_ of 21.5 ± 7.8 μg.h/mL on day 1 of treatment for RIF would be associated with DILI in 18–65 year old patients enrolled for the study, whilst a C_max_ of 5.9 ± 5.17 μg/mL or an AUC of 11.37 ± 10.57 μg.h/mL on day 1 of treatment for RIF would not be associated with DILI. INH levels of 3.03± 5.17 μg/mL or AUC_0–4h_ of 6.80 ± 5.96 μg.h/mL would be considered safe [85].

#### 4.3.3. Predicted Safety of INH in Children in Terms of DILI

Overall, the percentage of patients anticipated to be above the INH AUC DILI threshold of 21.78 μg.h/mL is 28% (out of 600 pediatric and young adult subjects simulated). This is very close to the value of 20% reported by Zheng et al. in an adult population [69]. However, the risk of DILI observation should be made considering the frequency of the genotypes in the target populations. Using the risk of DILI observation of 32.4% above the threshold and 9% below the threshold reported by Zheng et al. in an adult population [69], the following calculations can be made using the three scenarios for slow acetylator prevalence in the Brazilian population, as shown for example in Table 9.

The overall risk of developing a DILI based on INH exposure alone and depending on the prevalence of acetylator types in the Brazilian population would range from 13.5% to 18.7%, which is close to the value of 13.7% reported by Zheng et al. in an adult Chinese population [69]. Since the Chinese population comprises only 15% slow acetylators, it is expected that the incidence of DILI should be less than in the Brazilian population. Replacing the values reported for SA = 0.15, IA = 0.39, and RA = 0.46 in the Chinese population by Keller et al. [76] in Table 9, for example, would give a prediction of 12.6% incidence of DILI, which is very close to the measured values by Zheng et al. [69], further validating the model. As described previously, this DILI risk should be viewed as the risk for abnormal liver function tests in the treated population, but not as an organ failure. In order to assess the predictive ability of this model and conclusions, we compared the predicted incidence of abnormal liver function to the safety data reported in the literature for children.

Diallo et al. monitored the safety of a 9-month daily treatment of 10–15 mg/kg/day isoniazid in 407 children aged 0–17 years [81]. Over the treatment duration, only 6% of the children presented with minor symptoms which may have been related to the treatment and no children presented with serious adverse events that could have been related to the treatment. Donald provided a comprehensive review of antituberculosis drug-induced hepatotoxicity in children treated for tuberculosis or prophylactically treated with INH in combination or not with other drugs [78]. Overall, the results presented by Donald show that whether children are treated prophylactically or when TB is present, the presence of abnormal liver function results is more frequent with INH compared to RIF treatment. The percent of abnormal liver markers reported by Donald as a function of INH dose in children treated prophylactically or with active pulmonary TB is shown in Figure 7, together with the predictions for DILI risk in Table 9, which also correspond to elevated markers of liver function as per the Zheng et al. definition [69]. The prediction of elevated liver markers is comparable to observations reported in the literature at the 10 mg/kg INH dose. In addition, for pure phenotypes, the incidence of elevated biomarkers is close to that reported in the literature for children. For RA children, the risk calculated based on the modeling presented in this work of elevated liver markers is 9%, 26% for SA, and 12% for IA. In a study conducted in 54 children reported by Mantero et al. [78], phenotyped as RA for 22 children and SA for 32 children, out of 14 children developing increased AST levels, 2 were RA and 12 SA. For SA children, the modeling presented in this work shows a risk of abnormal liver markers at 26.1% when it is reported at 37.5% in the above study. For RA children, the modeling presented in this work shows a risk of abnormal liver markers at 9% when it is reported at 9.1% in the above study. The pure genotype risks and average risks are therefore considered comparable.

The percent of jaundice in children treated prophylactically with INH or treated with INH and other drugs for active pulmonary TB is low at most doses up to 15 mg/kg INH. For meningitis TB, there is around 10% of cases of jaundice from 10 to 20 mg/kg [78].

The conclusions of this analysis are shown below:

Prophylaxis with INH: For INH TB prophylaxis, children treated with INH alone from 4 to 20 mg/kg rarely develop jaundice with 2 cases reported for 22,256 treated subjects, which is a 0.009% incidence, whilst the presence of abnormal liver function tests will be 10.4% (196 abnormal results over 1892 tests). The incidence of signs of liver toxicity are relatively small at 0.629% (140 signs over 22,256 cases). There does not seem to be a dose effect in this evolution.

Pulmonary TB: In children with pulmonary TB treated with INH and other combination drugs with INH doses ranging from 5 to 20 mg/kg, there were overall 150 cases of jaundice reported for 17968 children treated. The overall incidence is 0.84%, which is significantly higher than in the prophylaxis group. The presence of abnormal liver function tests is reported in 9.9% of the cases (809 abnormal results over 8153 tests), which is very comparable with the prophylaxis group. There is a weak correlation with dose for the percent of abnormal liver function, but the reported data do not allow the level of detail to build a strong relationship against exposure. Donald reported a study by Mantero et al. who phenotyped 54 children as rapid (22) or slow (32) INH acetylators. Of the 14 children developing increased aspartate aminotransferase (AST) levels, only 2 (9.1%) were rapid INH acetylators and 12 (37.5%) were slow INH acetylators. These reported values are close to the ones calculated based on predicted exposure in children and DILI thresholds, such as in Table 9 where pure genotype risks of abnormal levels in liver function biomarkers are 9% in the RA group and 26.1% in the SA groups. The incidence of signs of liver toxicity is relatively small with 0.42% (76 signs over 17,968 cases). There does not seem to be a dose effect in this evolution. This percentage is also comparable to that observed for children under prophylactic treatment.

Meningitis TB: In children with meningitis TB treated with INH, RIF, and other agents at an INH dose from 10 to 20 mg/kg, the incidence of jaundice is more frequent with 144 reported cases out of 1434 patients, i.e., an incidence of 10%. In addition, the presence of abnormal liver function tests is reported in 52.9% of the tests. However, in the data reported by Donald, there were no reported signs of hepatotoxicity [78]. For these children who may have other organ failures and additional drugs in their treatment, it is hard to distinguish the impact of INH alone on liver toxicity, and it is concluded that the illness and co-administered drugs may potentialize the effect of chemotherapy on liver function.

Overall, when INH is used alone or in combination with RIF for the prophylaxis or active treatment of pulmonary TB in children at doses of 4–20 mg/kg, 10% of the cases will result in abnormal liver function tests, with 0.4–0.6% showing signs of hepatotoxicity. The jaundice incidence will range from 0.01 to 0.8% and depend on the overall health of children. Children with meningitis TB treated with INH doses of 10–20 mg/kg in association with RIF and other drugs will develop jaundice in 10% of the cases and there will be the presence of abnormal liver function tests in 53% of the cases. Co-morbidities and co-administered drugs may be responsible for these elevated incidences. The prediction of elevated liver function biomarker (DILI), run during this work for children treated with dispersible tablets comprising INH and RIF with the dosing schedule of Table 1, is compatible with the safety observed in children as reported by Donald at equivalent doses [78].

#### 4.3.4. Predicted Safety of RIF in Children in Terms of DILI and AKI

Based on RIF exposure predicted in children after administration of the dispersible tablet according to the dosing schedule of Table 1, the risk of developing DILI is estimated at 1.3% (Table 10). Based on RIF expression, the risk of developing AKI is estimated at 9.2% (Table 11), which is less than the AKI risk reported by Zheng et al. of 13.7% [69].

Diallo et al. monitored the safety of a 4-month daily treatment of 10–20 mg/kg/day RIF in 422 children aged 0–17 years [81]. Over the treatment duration, only 6.6% of the children presented minor symptoms, which may have been related to the treatment, and no serious adverse events were reported as being related to the treatment. From the data reported by Donald, RIF administered prophylactically at 10 mg/kg in 182 children led to a 6% incidence of abnormal liver function markers, whilst there were no reported cases of jaundice or hepatotoxicity [78]. These data confirm the predictions based on RIF exposure and adult human threshold values

## 5. Conclusions

This article highlights the benefits of using PBPK/PBBM to inform the development of an age-appropriate formulation and dosing for pediatric patients. In addition, PBBM combined with PK–safety and PK–efficacy relationships can also serve to run risk evaluations. The PBBM integrated the mechanistic dissolution of drug products and utilized the predicted in vivo disintegration time to adjust the average gastric emptying time. The PBBM was validated for INH, Ac-INH, and RIF in adult and pediatric patients across a variety of clinical trials. Overall, the risk identified with failing to achieve bioequivalence between the dispersible and adult reference fixed dose combination tablets is linked to INH. There is an anticipated higher INH C_max_ in adults and children for the dispersible tablet compared to the fixed dose combination, since the dispersible tablet will display faster gastric emptying time, and since INH belongs to BCS class 1, which makes it a marker of gastric emptying. Using this assumption for different gastric emptying and the ontogeny models available in GastroPlus, the exposure in children from birth to age 11 using the dispersible tablets was predicted for INH and RIF. For INH, on top of C_max_ differences related to the formulation, the genotypical acetylator status of patients was the main factor responsible for the differences in exposure.

Monitoring of the liver and kidney function in treated patients is recommended across all age groups, as the main risk involves elevated liver function biomarkers in approximately 15% of the patients due to INH exposure (53% for meningitis TB). However, the risk of developing jaundice is relatively low at 0.8% (10% for meningitis TB) and the incidence of actual signs of liver injury is also low (0.4–0.6%). AKI in children is anticipated to be similar to or less than those reported in the literature for adult patients. The modeling approach for the predicted elevation of liver function biomarkers is validated since it can reproduce the observed values in children. Exposures reached for INH and RIF at the recommended doses are anticipated to be effective in children of all age groups. Altogether, considering the sum of data available in the literature, the dissolution performance of the proposed dispersible tablet formulation, and the simulations conducted in the target populations, it is concluded that dispersible tablets are anticipated to be safe and effective to support the maintenance treatment of TB in children from birth to 10 years of age.

## Figures and Tables

**Figure 1 pharmaceutics-16-01587-f001:**
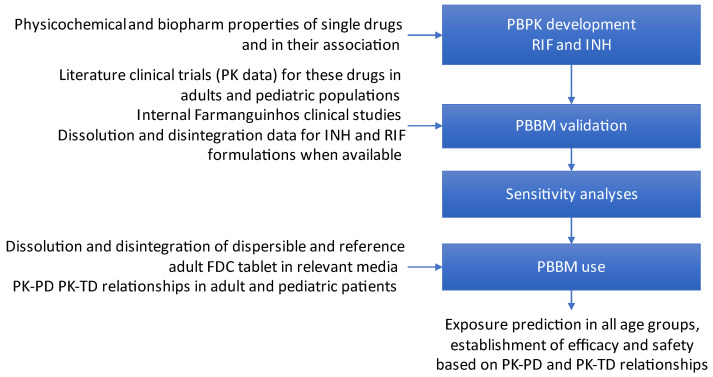
Overview of the modeling strategy.

**Figure 2 pharmaceutics-16-01587-f002:**
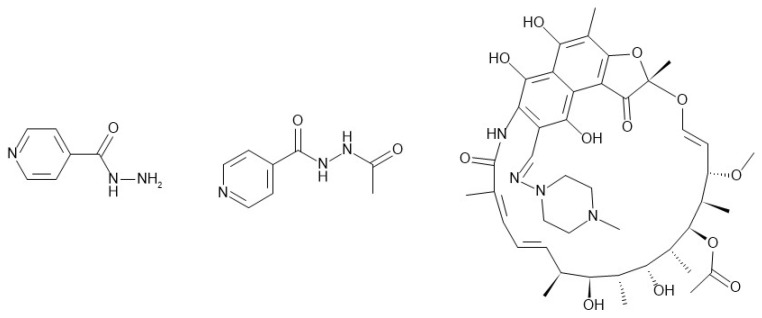
Structure of INH (**left**), Ac-INH (**middle**), and RIF (**right**).

**Figure 3 pharmaceutics-16-01587-f003:**
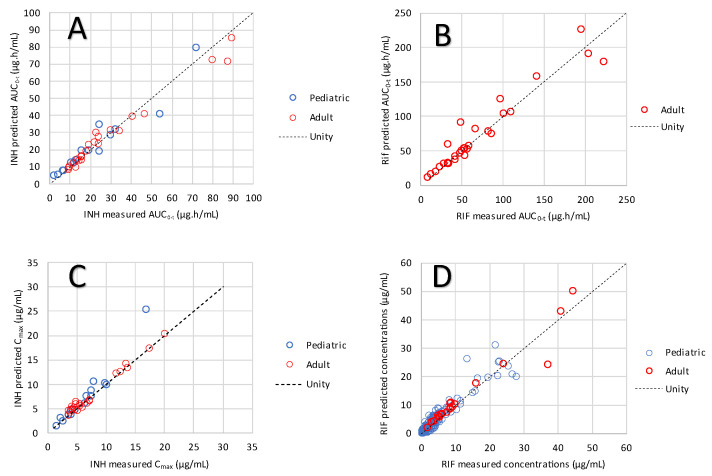
Prediction of INH AUC (**A**), RIF AUC (**B**), INH C_max_ (**C**), and RIF C_max_ and plasma concentrations (**D**) across all the validation clinical datasets for the adult and pediatric studies.

**Figure 4 pharmaceutics-16-01587-f004:**
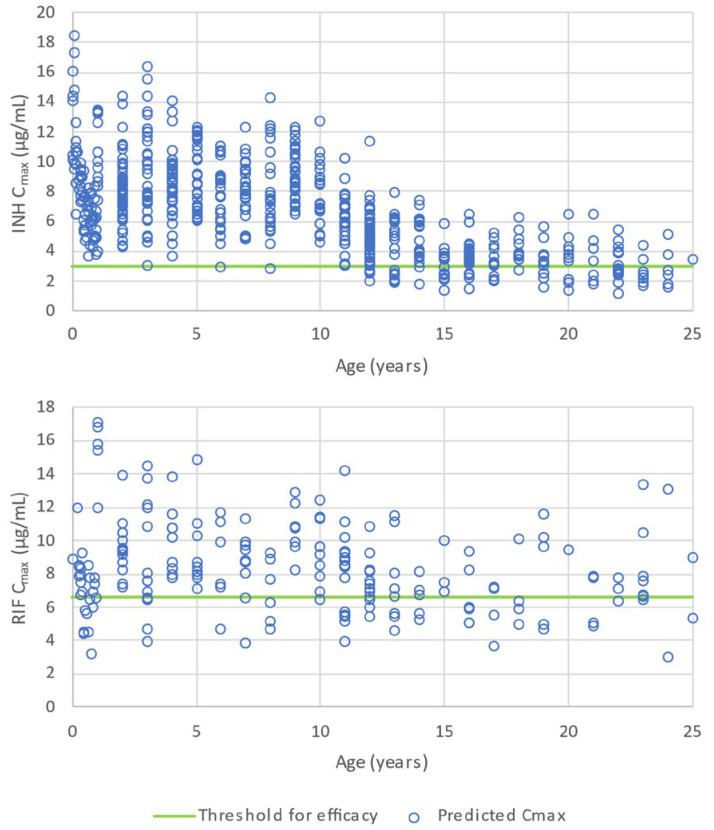
C_max_ predicted for populations of pediatric subjects for INH (**upper panel**) and RIF (**lower panel**) by age group according to the dosing schedule of Table 1. The horizontal line shows the minimum threshold for efficacy according to Kiser et al. [57] for INH (**upper panel**) and Pasipanodya et al. [68] for RIF (**lower panel**).

**Figure 5 pharmaceutics-16-01587-f005:**
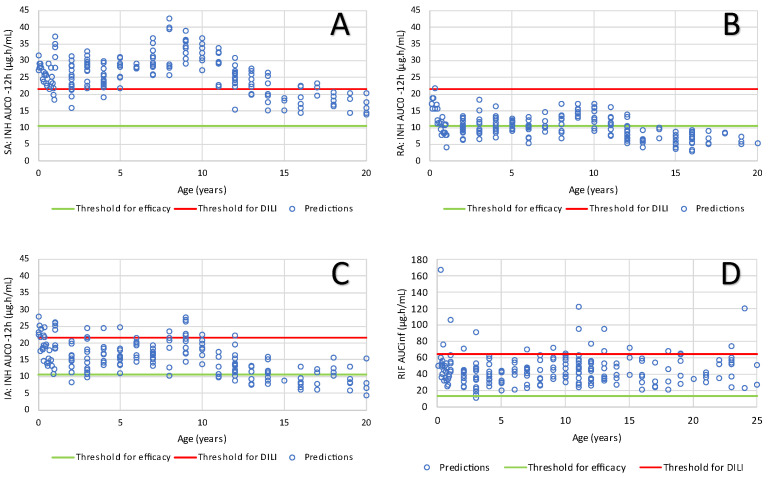
AUC predictions for pediatric populations of INH SA (**A**), INH RA (**B**), INH IA (**C**), and RIF (**D**) according to the schedule of Table 1. The horizontal lines show the minimum AUC for efficacy and maximum adult AUC for INH DILI according to Zheng et al. [69]. The horizontal green line for panel (**D**) shows the threshold for efficacy according to [68].

**Figure 6 pharmaceutics-16-01587-f006:**
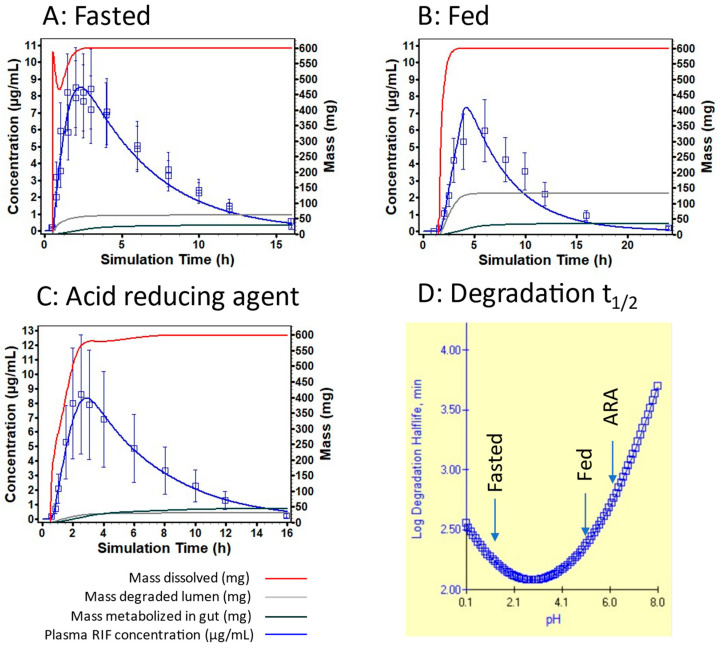
PK profile prediction for 600 mg RIF administered in the fasted state (**A**), fed state (**B**), and following ARAs (**C**). The PK data are reported by Peloquin et al. [80]. Panel (**D**) shows the log degradation half-life for RIF in the fasted state, fed state, and following ARA administration.

**Figure 7 pharmaceutics-16-01587-f007:**
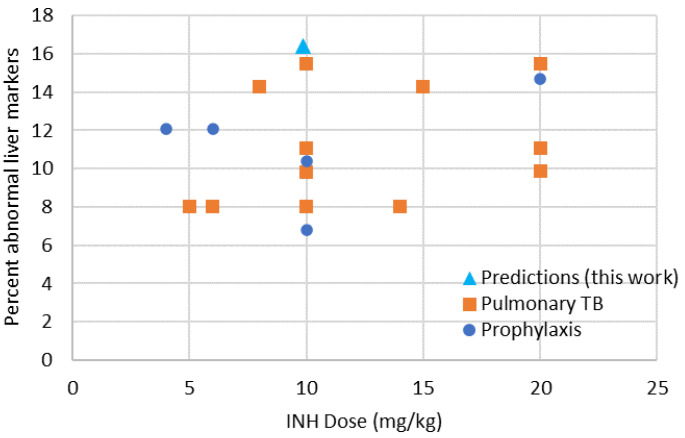
Evolution of percent abnormal liver markers in children treated prophylactically or with active pulmonary TB as a function of INH dose from Donald [78], compared to predictions resulting from this work using the average Brazil genotype reported in [86] and risk exposure thresholds reported by Zheng et al. [69].

**Table 1 pharmaceutics-16-01587-t001:** Calculation of posology for children based on dosing recommendations for each age group.

Age (Years)	Child Weight (kg)	Dose RIF (mg)	Dose INH (mg)	Number of Dispersible Tablets
0	4.6	75	50	1
1	9.2	150	100	2
2	12	150	100	2
3	14	225	150	3
4	16	225	150	3
5	18	300	200	4
6	21	300	200	4
7	23	300	200	4
8	26	450	300	6
9	30	450	300	6
10	34	450	300	6
11	39	600	300	Adult tablet

**Table 2 pharmaceutics-16-01587-t002:** Input parameters used for isoniazid PBPK model development.

Parameter (Unit)	Value for INH	Rationale/Reference(s)
1. Physicochemical and Binding Properties		
Molecular mass (g/mol)	137.14	From structure
Type of drug substance	Crystalline	[34]
Log P	0.64	[35]
pKa	2.13(B), 3.81(B), 11.03(A)	Measured [35]
Intrinsic solubility (mg/mL)	161	[34]
Human blood-to-plasma ratio (Rbp)	1.158	AP v10.3
f_u, plasma_ (%)	84	APv10.3
2. Absorption		
Human effective jejunal permeability (P_eff_) (×10^−4^ cm/s)	6.7	Calculated from ref. [36]
3. Distribution		
Method	Full body PBPK, Lukacova method of Kp prediction for all tissues	GastroPlus default method
4. Metabolism		
K_m,u_ NAT2 (mg/L)	5	Fitted on INH profiles at different doses
V_max_ NAT2 gut (mg/s), RA	4.3 × 10^−2^	Scaled from liver V_max_ values
V_max_ NAT2 gut (mg/s), IA	2.72 × 10^−2^	Scaled from liver V_max_ values
V_max_ NAT2 gut (mg/s), SA	1.15 × 10^−2^	Scaled from liver V_max_ values
V_max_ NAT2 PBPK (mg/s/mg-enz), RA	4.94 × 10^−3^	Fitted to subject B from ref. [37]
V_max_ NAT2 PBPK (mg/s/mg-enz), IA	3.13 × 10^−3^	Average of RA and SA values
V_max_ NAT2 PBPK (mg/s/mg-enz), SA	1.32 × 10^−3^	Fitted to subject B from ref. [37]
AcINH/INH	1	From structure
5. Elimination		
CL_R_ (L/h/kg)	Given by f_u,p_ × GFR	Default

GFR: glomerular filtration rate, RA: rapid acetylator, IA: intermediate acetylator, SA: slow acetylator, f_u,p_: unbound drug fraction in plasma.

**Table 3 pharmaceutics-16-01587-t003:** Input parameters used for acetyl-isoniazid PBPK model development.

Parameter (Unit)	Value for INH	Rationale/Reference(s)
1. Physicochemical and Binding Properties		
Molecular mass (g/mol)	179.18	APv10.3
Log P	−0.35	APv10.3
pKa	3.26(B), 9.27(A), 10.37(A)	APv10.3
Intrinsic solubility (mg/mL)	3.73	APv10.3
Human blood-to-plasma ratio (R_bp_)	0.86	APv10.3
F_u,plasma_ (%)	80.94	APv10.3
2. Absorption		
Human effective jejunal permeability (P_eff_) (×10^−4^ cm/s)	2.5	APv10.3
3. Distribution		
Method	Full body PBPK, Lukacova method of Kp prediction for all tissues	GastroPlus default method
4. Metabolism		
V_max_, 1A2 (mg/s/mg-enz)	1.288 × 10^−3^	Fitted to data from Bing et al. [38]
K_m,u_ 1A2 (mg/L)	115.4	AP v10.3
5. Elimination		
CL_R_ (L/h/kg)	Given by f_u,p_ × GFR	Default

GFR: glomerular filtration rate, f_u,p_: unbound drug fraction in plasma.

**Table 4 pharmaceutics-16-01587-t004:** Input parameters used for rifampicin PBPK model development.

Parameter (Unit)	Value for INH	Rationale/Reference(s)
1. Physicochemical and Binding Properties		
Molecular mass (g/mol)	822.96	From structure
Log P	1.5	APv10.3 predicted 2.528, close to value reported by Ermondi [39]
pKa	2.97 (A), 7.5 (B)	Measured from [39]
Aqueous solubility (mg/mL)	0.64 @ pH 5.5	Measured from [40]
Human blood-to-plasma ratio (Rbp)	0.738	APv10.3
f_u, plasma_ (%)	13.92 (human)	From [41] not far from 17.4% from APv10.3 or 13.3% measured in [42]
2. Absorption		
Human effective jejunal permeability (P_eff_) (×10^−4^ cm/s)	2.11	Scaled from Caco2 data from Biganzoli et al. [43]
3. Distribution		
Method	Full body PBPK, Lukacova method of Kp prediction for all tissues	Fitted to IV data from Wasserman et al. [44]
4. Metabolism		
K_m,u_ CYP3A4 (mg/L)	14.11	APv10.3 calculated from APv10.3 value of 17.153 μM
V_max_ CYP3A4 gut (mg/s)	1.4 × 10^−2^	Fitted to oral PK data from Loos [45]
V_max_ CYP3A4 PBPK (mg/s/mg-enz)	4.06 × 10^−4^	Fitted to IV data from Wasserman et al. [44]
K_m,u_ CES2 (mg/L)	14.11	Same as CYP3A4
V_max_ CES2 PBPK (mg/s/mg-enz)	2.61 × 10^−4^	Fitted to IV data from Wasserman et al. [44]
V_max_ CES2 gut (mg/s)	1.4 × 10^−2^	Fitted to oral PK data from Loos [45]
5. Elimination		
CL_R_ (L/h/kg)	Given by f_u,p_ × GFR	Default

GFR: glomerular filtration rate, RA: rapid acetylator, IA: intermediate acetylator, SA: slow acetylator, f_u,p_: unbound drug fraction in plasma.

**Table 5 pharmaceutics-16-01587-t005:** Calculation of performance indicators for model validation.

Analyte	Population	PK Parameter	PE (%)	AFE
INH	Adult	AUC	3.8	0.97
INH	Pediatric	AUC	9.4	1.09
INH	Adult	Cmax	5.1	1.03
INH	Pediatric	Cmax	7.3	1.08
RIF	Adult	AUC	11.7	1.10
RIF	Adult	Cmax	10	1.15
RIF	Pediatric	Plasma concentrations	28.5	1.14

**Table 6 pharmaceutics-16-01587-t006:** Virtual cross-over studies between pediatric dispersible formulation (test) and adult fixed dose combination (reference) at 300 mg INH + 600 mg RIF.

Population	Analyte	PK Parameter	GMR	90% CI	VBE Conclusion
SA	INH	AUC_inf_	104.3	99.0–109.8	Passed
IA	INH	AUC_inf_	108.9	100–118.7	Passed
RA	INH	AUC_inf_	103.3	94.3–113.2	Passed
SA	INH	C_max_	111.7	105.6–118.1	Passed
IA	INH	C_max_	115.4	106.6–124.9	Passed
RA	INH	C_max_	108.4	98.5–119.2	Passed
-	RIF	AUC_inf_	99.5	87.3–113.4	Passed
-	RIF	C_max_	102	91.4–113.9	Passed

CI: confidence interval, GMR: geometric mean ratio, VBE: virtual bioequivalence.

**Table 7 pharmaceutics-16-01587-t007:** Probability of positive 2-month sputum culture based on RIF C_max_.

Parameter	Description	Value
A	Probability to be over C_max_ threshold 6.6 μg/mL ^1^	0.8
B	Probability of positive 2-month sputum culture above threshold ^2^	0.01
C	Probability of positive 2-month sputum culture under threshold ^2^	0.19
D	Overall probability of positive 2-month sputum culture ^3^	0.046

^1^: Based on simulations. ^2^: Probabilities taken from Pasipanodya [68]. ^3^: $D=A\times B+\left(1-A\right)\times C.$

**Table 8 pharmaceutics-16-01587-t008:** Probability of poor 2-year therapy outcome based on RIF exposure.

Parameter	Description	Value
A	Probability to be over AUC threshold 13 μg.h/mL ^1^	0.95
B	Probability of poor therapy outcome above threshold ^2^	0.12
C	Probability of poor therapy outcome under threshold ^2^	0.33
D	Overall probability of poor therapy outcome ^3^	0.131

^1^: Based on simulations. ^2^: Probabilities taken from Pasipanodya [68]. ^3^: $D=A\times B+\left(1-A\right)\times C$.

**Table 9 pharmaceutics-16-01587-t009:** DILI risk evaluation for Brazil based on INH exposure.

Parameter	Description	Value for Brazil		
A	Genotype	SA	IA	RA
B	Frequency of genotype ^1^	0.34	0.55	0.11
C	Probability to be over AUC threshold 21.78 μg.h/mL ^2^	0.73	0.12	0
D	Probability of DILI above threshold ^3^	0.324	0.324	0.324
E	Probability of DILI under threshold ^3^	0.09	0.09	0.09
F	Overall DILI in pure genotype ^4^	0.261	0.118	0.09
G	Overall DILI risk in population ^5^	0.164		

^1^: From [86]. ^2^: From simulation results. ^3^: Probabilities taken from [69]. ^4^: $F=C\times D+\left(1-C\right)\times E$. ^5^: $G=F_{SA}B_{SA}+F_{IA}B_{IA}+F_{RA}B_{RA}$.

**Table 10 pharmaceutics-16-01587-t010:** DILI risk evaluation based on RIF exposure.

Parameter	Description	Value
A	Probability to be over AUC threshold 64.49 μg.h/mL ^1^	0.057
B	Probability of DILI above threshold ^2^	0.155
C	Probability of DILI under threshold ^2^	0.0039
D	Overall DILI risk in population ^3^	0.013

^1^: Based on simulations. ^2^: Probabilities taken from Zheng et al. [69]. ^3^: $D=A\times B+\left(1-A\right)\times C.$

**Table 11 pharmaceutics-16-01587-t011:** AKI risk evaluation based on RIF exposure.

Parameter	Description	Value
A	Probability to be over AUC threshold 82.01 μg.h/mL ^1^	0.023
B	Probability of AKI above threshold ^2^	0.579
C	Probability of AKI under threshold ^2^	0.081
D	Overall AKI risk in population ^3^	0.092

^1^: Based on simulations. ^2^: Probabilities taken from Zheng et al. [69]. ^3^: $D=A\times B+\left(1-A\right)\times C.$

## Data Availability

Data supporting the reported results can be found in the article and Appendix A.

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
