# Peer review of "The Development of an Age-Appropriate Fixed Dose Combination for Tuberculosis Using Physiologically-Based Pharmacokinetic Modeling (PBBM) and Risk Assessment"

_pharmaceutics, 2024, doi:10.3390/pharmaceutics16121587_

Round 1
Reviewer 1 Report
Comments and Suggestions for Authors
The article “The development of age-appropriate fixed dose combination for tuberculosis using physiologically-based pharmacokinetic modeling (PBBM) and risk assessment” is an interesting piece of work. I have the following comments/suggestions,
1. I think it will be more interesting for the readers if the authors can the name of modeling and simulation platform used for the work in the abstract.
2. The introduction section does not contain any information regarding the already published PBPK models on these drugs, it will be more interesting for the readers to see what is already available in literature and what this model adds.
3. The method section is well described.
4. Figure 3, it is very difficult to follow the data as it is unreadable. In part C one can hardly differentiate between the lines.
5. Some references are not linked as on line 442 and 446 and other places it is showing "Error!Reference not found."
6. Figure 6, the authors should make the legends clear, as they are confusing.
Author Response
Comment 1: I think it will be more interesting for the readers if the authors can the name of modeling and simulation platform used for the work in the abstract.
Response 1: Thanks for this comment, the abstract was updated accordingly.
Comment 2: The introduction section does not contain any information regarding the already published PBPK models on these drugs, it will be more interesting for the readers to see what is already available in literature and what this model adds.
Response 2: Thanks for this comment, we have updated the introduction to detail pre-existing literature on this topic.
Comment 3: The method section is well described.
Response 3: Thanks for this comment
Comment 4: Figure 3, it is very difficult to follow the data as it is unreadable. In part C one can hardly differentiate between the lines.
Response 4: We agree with the reviewer’s comment. We have removed this figure and provided a table instead which shows the GMR and 90% confidence interval and concludes on the bioequivalence for these comparisons.
Comment 5: Some references are not linked as on line 442 and 446 and other places it is showing "Error!Reference not found."
Response 5: Apologies for that. The links to the tables broke during compilation. We have updated the links
Comment 6: Figure 6, the authors should make the legends clear, as they are confusing.
Response 6: We agree with the reviewer and have updated Figure 6 with a clearer legend
Reviewer 2 Report
Comments and Suggestions for Authors
Comments:
This manuscript developed and validated a PBBM model to predict the exposure of INH and RIF from the age-appropriate formulation and adult tablet in different age groups from birth to adulthood. The authors did a lot of work and the method is sound.
A few comments:
1. There are big differences among children within I year old. For example, neonates are defined in the 0-1 year group, but they may show different PK profiles. For example, in Figure 4, high variability can be seen at the INH concentrations for children within 1 year. Please provide some information for the reason.
2. There are al lot of reference error messages; please double-check. The reference format on line 315 is different from others; please correct it and keep it consistent.
3. In Tables 2,3 and 4, what the “B” means should also be explained. The format of data exhibition in these tables should be consistent.
4. The PK results in Figure 3 should be removed and summarized in one table and moved to supplementary material.
Author Response
Comment 1: There are big differences among children within I year old. For example, neonates are defined in the 0-1 year group, but they may show different PK profiles. For example, in Figure 4, high variability can be seen at the INH concentrations for children within 1 year. Please provide some information for the reason.
Response 1: Thanks for the question, we have explained the main drivers for this variability in the updated discussion of the manuscript.
Comment 2: There are a lot of reference error messages; please double-check. The reference format on line 315 is different from others; please correct it and keep it consistent.
Response 2: Apologies for that. The links to the tables broke during compilation. We corrected the links
Comment 3: In Tables 2,3 and 4, what the “B” means should also be explained. The format of data exhibition in these tables should be consistent.
Response 3: Thanks for this question, we have updated all the footnotes and ensured consistency across these tables.
Comment 4: The PK results in Figure 3 should be removed and summarized in one table and moved to supplementary material.
Response 4: We agree with the reviewer and have created a table to summarize the outcome of the virtual BE trials.
Round 2
Reviewer 2 Report
Comments and Suggestions for Authors
Thank you for your response. I am satisfied with this revision.